# LinguaPhylo: A probabilistic model specification language for reproducible phylogenetic analyses

**Alexei J. Drummond**[1,2,3]*, **Kylie Chen**[1,2,3], **Fábio K. Mendes**[1,4], **Dong Xie**[1,2,3]

1 Centre for Computational Evolution, University of Auckland, Auckland, New Zealand, 2 School of Biological Sciences, University of Auckland, Auckland, New Zealand, 3 School of Computer Science, University of Auckland, Auckland, New Zealand, 4 Department of Biology, Washington University in St. Louis, St. Louis, United States of America

* a.drummond@auckland.ac.nz

**Data Availability Statement:** All data analysed is available in the public GitHub repository https://github.com/LinguaPhylo/linguaPhylo.

## Abstract

Phylogenetic models have become increasingly complex, and phylogenetic data sets have expanded in both size and richness. However, current inference tools lack a model specification language that can concisely describe a complete phylogenetic analysis while remaining independent of implementation details. We introduce a new lightweight and concise model specification language, 'LPhy', which is designed to be both human and machine-readable. A graphical user interface accompanies 'LPhy', allowing users to build models, simulate data, and create natural language narratives describing the models. These narratives can serve as the foundation for manuscript method sections. Additionally, we present a command-line interface for converting LPhy-specified models into analysis specification files (in XML format) compatible with the BEAST2 software platform. Collectively, these tools aim to enhance the clarity of descriptions and reporting of probabilistic models in phylogenetic studies, ultimately promoting reproducibility of results.

This is a *PLOS Computational Biology* Software paper.

## 1 Introduction

Transparency is a scientific ideal, and replicability and reproducibility lie at the heart of the scientific endeavour [1, 2]. Metaresearch efforts have uncovered the so-called 'reproducibility crisis' [3] in many scientific domains [3]. In recent years, the growing number of computational biology software packages available has enabled greater choice in data analyses, but at the cost of increased complexity in the data-preparation and analytical pipelines [4]. This increases the difficulty of accurately reporting and reproducing analyses. These barriers have been recognised by the wider genomics research community [4] as well as within evolutionary biology [5].

**Funding:** AJD was supported by a James Cook Fellowship (JCF-UOA1901) from the Royal Society of New Zealand (https://www.royalsociety.org.nz). FKM was supported by Marsden grant 16-UOA-277 from the Royal Society of New Zealand and by National Science Foundation (https://www.nsf.gov) grant DEB-2040347. These funders played no role in the study design, data collection, analysis, decision to publish or preparation of the manuscript.

**Competing interests:** The authors have declared that no competing interests exist.

In evolutionary biology, phylogenetics has become a highly technical discipline [5]. The most general phylogenetic tools are Bayesian methods including BEAST [6], BEAST 2 [7, 8], MrBayes [9] and RevBayes [10] that can simultaneously reconstruct phylogenetic tree topology and divergence times, as well as estimate the related micro-evolutionary and macro-evolutionary parameters. Phylogenetic analyses often combine multiple models within a complex pipeline to answer questions in evolutionary biology such as species evolution [11, 12], ancestral biogeographical ranges [13, 14], and epidemic dynamics [15, 16].

Reproducing, reusing and interpreting a phylogenetic model is not trivial, and requires an understanding of the input data, details of the model (i.e., its parameters and how they are related and their priors), and inference methodology. The latter can include complex Markov chain Monte Carlo (MCMC) proposal distributions and sampling algorithms which are not part of the model. Currently, little research has been done on the readability, reproducibility and reusability of phylogenetic analyses employing phylogenetic models. Our paper presents a tool that aims to: (i) facilitate concise and exact communication of phylogenetic models, (ii) improve reproducibility, and (iii) increase reusability of phylogenetic models and their variations on new datasets.

Probabilistic graphical models (PGM) have previously been introduced to phylogenetic inference by Höhna et al. [10], where they are described in the Rev language of RevBayes [10]. Previous attempts to address model specification of Bayesian phylogenetic analyses include BEAST-style XMLs (eXtensible Markup Language) developed for the BEAST software [6–8], the Nexus-based language of MrBayes [9] and the aforementioned Rev programming language used in RevBayes [10]. The extensibility of XMLs provides flexibility to developers allowing them to create new descriptive tags for specifying new models. Unfortunately, BEAST XMLs can be hard to read due to their XML-based, verbose syntax, which is unfamiliar to many users. The deeply nested structures and hierarchical organisation in BEAST XMLs make understanding model components and their relationships more challenging. In contrast, more general probabilistic programming languages such as found in JAGS [17], BUGS [18, 19] and Stan [20] employ more concise, linear structures that simplify comprehension and modification of models. Additionally, probabilistic programming languages are more accessible and widely applicable within the statistical modelling community, while BEAST XMLs are more domain-specific. The Rev language [10] offers an alternative to XMLs, incorporating conventional notation from general probabilistic programming languages, making model specifications more recognizable and flexible for statistically literate users. However, users still face challenges with verbose and extraneous implementation details, such as MCMC sampling settings, logging information, and proposal distributions. Additionally, the task of accurately describing the Rev model in natural language for a manuscript's methods section can be error-prone, further complicating the process of model specification and communication.

In this study, we present LinguaPhylo (LPhy, which we pronounce 'el-fee'), an open-source model specification language designed to enhance the readability, reproducibility, and reusability of phylogenetic models. LPhy boasts a simple syntax that enables succinct specification of complex models, and is implemented within a framework that generates accurate textual descriptions and graphical diagrams of phylogenetic models based on user input.

## 2 Design and implementation

The LPhy language is designed to enable the specification of phylogenetic models using a concise and readable syntax. The reference implementation is built on top of the Java

programming language and provides features for: (i) concise formal specification of phylogenetic models on real or synthetic data, (ii) data simulation from phylogenetic models, (iii) integration with the BEAST 2 phylogenetic inference framework, and (iv) an extensibility mechanism for adding new functionality and data types to the LPhy language.

## 2.1 Language features

The LPhy language is described by an Extended Backus-Naur form grammar (EBNF) [21]. We used ANTLR [22, 23] to generate an LPhy parser in Java. This Java-based LPhy parser was used as the foundation for the development of LPhyStudio and LPhyBEAST software packages. The ANTLR parser generator can also be targeted to other general programming languages like Python, C++ and Javascript. This gives a relatively easy path to implementation of LPhy support in other phylogenetic software packages.

The main components of the LPhy language grammar are variables, arrays of variables, and generators. There are three classes of generators: (i) generative distributions which produce values for random variables, (ii) deterministic functions that produce the deterministic values given the same input values, and (iii) method calls. For deterministic functions and method calls, these generators produce deterministic nodes in the PGM. This is illustrated in Fig 1, which shows a graphical representation of the model specified in Example 1. Deterministic nodes are shown as diamonds (e.g., the 'Q' matrix of the Jukes-Cantor model [24]). Stochastic nodes are represented by circles (e.g., Θ, the population size governing the coalescent times generated by the Coalescent process [25]), and constant nodes are represented by squares (e.g., the mean of the log-normal generative distribution underlying Θ).

An LPhy script is a text file with the '.lphy' file extension and is case-sensitive. In the reference implementation, syntax checking is performed during execution. Control flow structures are not allowed in order to promote simplicity and readability, facilitating a lower barrier to entry and a gentler learning curve. Instead of loop structures, we provide implicit vectorization

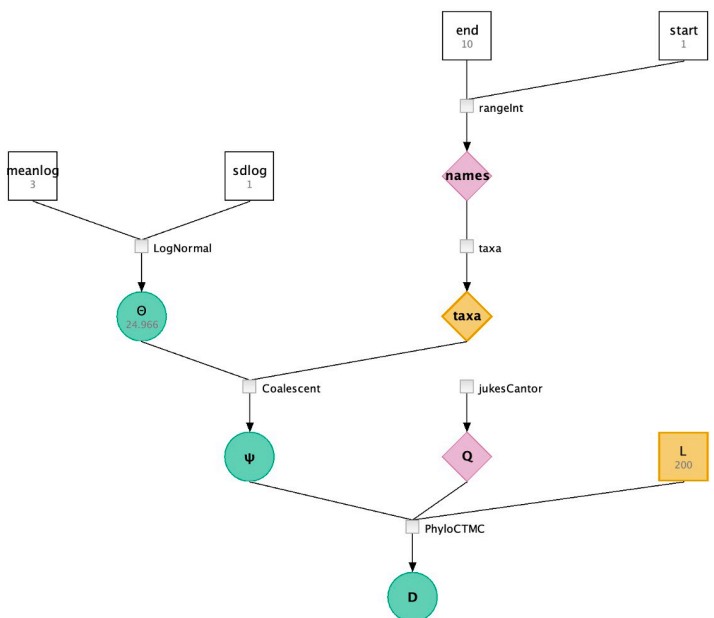

**Fig 1. The graphical representation of the probabilistic model defined in Example 1.**

similar to R [26]. Allowing all arguments of distributions and functions to be vectorised results in more compact and expressive model specifications. This can lead to clearer representations of the model's structure and relationships. Additionally an optional 'replicates' argument allows users to generate vectors of independent and identically distributed (IID) random variables easily, further simplifying the model specification. When designing new generators, optional arguments in functions and generative distributions are allowed, these arguments may or may not have default values. Two distinct code blocks are used to differentiate between the part of the script describing the data (the data block), and part describing the model (the model block). The code block structure is inspired by the programming language Stan [20]. The syntax for the specification of random variables is similar to probabilistic programming languages such as JAGS [17], BUGS [18, 19] and Stan [20]. Scripts can be loaded into LPhyStudio via the menu or the toolbar. Additionally, we provide a console for executing LPhy commands line by line within LPhyStudio.

The LPhy language grammar does not specify any explicit types, nor does it specify any protected pre-defined generative distributions or functions, except for a very small number of mathematical functions that allow simple expressions. Accompanying the LPhy language grammar is the LPhy reference implementation, a Java implementation offering standard statistical and phylogenetic distributions, as well as supporting functions with specified Java types. Implementers of the LPhy language in other systems should support the reference distributions, using equivalent types in their respective languages.

In the reference Java implementation, variables can possess primitive or custom types. Primitive types include doubles, integers, booleans, and strings, while custom types, such as alignments, trees, and discrete traits, are internally created as Java Objects by invoking the generator's constructor. Variables can be vectorised into an array of elements using the 'replicates' argument. Java-style overloading supports function overloading, and type checking for generator arguments is performed during execution.

**2.1.1 Syntax.** In LPhy, each line's syntax consists of a variable declaration on the left-hand side, a specification operator, and a generator on the right-hand side, with lines ending with a semicolon character. For instance:

```
b ~ Normal(mean= 0.0, sd=1.0);
```

In this example, variable $b$ is specified by the normal generative distribution `Normal()` with two arguments: the mean `mean` (0.0) and the standard deviation `sd` (1.0).

The left-hand side declares the name of a variable or an array of variables (case sensitive). The right-hand side specifies the values of the variable or array. This can be a constant value, array of constant values, deterministic function or stochastic generative distribution. Deterministic functions and generative distributions are matched by method signatures from constructors of their corresponding Java class in the reference implementation. See the LPhy reference implementation manual available from the homepage https://linguaphylo.github.io/ for a complete list of functions and generative distributions. Arguments inside functions or generative distributions follow the convention: (*argument name*) = (*value*).

**2.1.2 Specification operators.** An equal sign = is used to specify deterministic or constant values for variables, such as:

```
a = 2.0;
```

A tilde sign ∼ is a specification operator which denotes the relationship between a stochastic random variable and its generator. In a Bayesian context, this is the prior distribution of the

random variable. Or when given observed data, this specifies the likelihood function. For example, this specifies a prior for the variable *b*:

```
b ~ Normal(mean=0.0, sd=1.0);
```

The type of a variable is inferred from the return type of its generator and does not need to be declared. For arguments of functions or generative distributions, the types are defined in the LPhy reference implementation manual available from the homepage https://linguaphylo. github.io/.

**2.1.3 Arrays.**   Arrays can be defined using square brackets with elements delimited by comma separators. For sequences of consecutive integers, we allow a more compact notation using the colon ':' to define a range. For example:

```
c = [1, 2, 3, 4, 5];
d = 2:10;
```

The variable c has an array with values (1, 2, 3, 4, 5). The variable d has an array with values 2 to 10.

**2.1.4 Code blocks.**   The **data** and **model** keywords are reserved to specify code blocks inside curly brackets. The data block is used to read in and store input data, which are used by the model. Within the data block, we can read in alignment data via the NEXUS or FASTA parsers, specify constant values, and store metadata about the dataset. The model block is used to define the models and parameters in a Bayesian phylogenetic analysis.

```
data {
  L = 200;
  taxa = taxa(names=1:10);
}
model {
  Θ ~ LogNormal(meanlog=3.0, sdlog=1.0);
  ψ ~ Coalescent(theta=Θ, taxa=taxa);
  D ~ PhyloCTMC(L=L, Q=jukesCantor(), tree=ψ);
}
```

**Example 1**. An LPhy script defining a constant-size coalescent tree prior with log-normally distributed population sizes, a strict clock model, and a Jukes-Cantor model on 10 nucleotide sequences with 200 sites (base pairs).

Example 1 specifies a complete phylogenetic model using only five lines of code inside two blocks. The first block specifies '200' nucleotide sites in 'L' and ten taxa named from 1 to 10 in 'taxa'. Taxa can be declared as strings or as numbers. In this example, the taxa names are numbered. The second block declares stochastic nodes or random variables highlighted in bluish-green, and their generative distributions highlighted in blue. Constant nodes with fixed values are shown in vermillion.

Fig 1 shows this model specification represented as a probabilistic graphical model. Stochastic nodes are shown as circles, deterministic functions are shown as diamonds, and constants are shown as squares.

**2.1.5 Variable vectorization.**   Named variables can be scalars or vectors. Any generator can be vectorized to produce a vector of IID random variables using the `replicates` keyword:

```
κ ~ LogNormal(meanlog=0.5, sdlog=1.0, replicates=3);
```

In the example above, $k$ is a random vector of three log-normally distributed IID values.

Vectorization can also be applied to a generative distribution that produces vectors. In this case, the output will be a matrix, as seen in the following example.

```
π ~ Dirichlet(conc=[2.0, 2.0, 2.0, 2.0], replicates=3);
```

Here, $\pi$ contains 3 vectors, where each vector represents nucleotide base frequencies. So the resultant matrix will be 3 x 4 (major dimension of 3 and a minor dimension of 4).

A second mechanism for vectorization can be used by passing a vector of elements instead of a single element as an input argument of a generator:

```
κ ~ LogNormal(meanlog=0.5, sdlog=1.0, replicates=3);
π ~ Dirichlet(conc=[2.0, 2.0, 2.0, 2.0], replicates=3);
Q = hky(kappa=κ, freq=π);
```

Here, $k$ and $\pi$ are vectors, both with a major dimension of 3. These are passed as arguments into the `hky` deterministic function, which returns a vector containing three instantaneous rate matrices stored in $Q$.

**2.1.6 Extension mechanism.** LPhy is designed to be modular and extensible to encourage integrative software development and adoption by method developers in the field. LPhy core and its extensions are built using Java 17 with long-term support (LTS) and Gradle 4. The LPhy extension mechanism is implemented using the Java Platform Module System (JPMS) and the Java Service Provider Interface (SPI). This modular extension framework allows developers greater flexibility in code development and software releases. New functionality such as generative distributions, data types, and deterministic functions can be developed within LPhy extension modules. Additionally, software releases can be done independently to core LPhy releases. An example of the LPhy extension mechanism is the Phylonco package [27] which is described in the results section.

**2.1.7 Parametric distributions.** The LPhy reference implementation comes with a series of parametric distributions commonly used in evolutionary models, including uniform, normal, log-normal, gamma, exponential, and dirichlet distributions. Parametric distributions can be specified as generative distributions for model parameters by:

```
μ ~ LogNormal(meanlog=-5.0, sdlog=1.25);
```

Each parametric distribution is characterized by its own parameters. In the example above, the extinction rate parameter $\mu$ is drawn from a log-normal distribution with mean -5 and standard deviation 1.25 in log space.

**2.1.8 Tree models.** Tree models are used to generate phylogenetic trees, and are central components in phylogenetic simulation and analysis. We briefly describe some of the main tree models implemented in LPhy below. This includes coalescent models and birth-death models.

**Serially sampled coalescent model**

The simplest coalescent model LPhy implements is the constant-population size coalescent, which can be extended to generate serially sampled (heterochronous) data [28]:

```
Θ ~ LogNormal(meanlog=3.0, sdlog=1.0);
ψ ~ Coalescent(theta=Θ, taxa=taxa(names=["a", "b", "c", "d"],
  ages=[0.0, 1.0, 2.0, 3.0]));
```

The script above specifies a serially sampled constant-population size coalescent (a generative distribution) for tree $\psi$ with four taxa, 'a', 'b', 'c', 'd' sampled at 0.0, 1.0, 2.0 and 3.0 time points, respectively. Here, sample time is defined as the age of a sample, where 0.0 represents the present moment.

**Structured coalescent model.** The structured coalescent [29, 30] generalizes the constant-population size coalescent [25] by allowing multiple demes, each of which are characterized by a distinct population size. In the simplest case this population size does not change through time. Demes exchange individuals according to migration rates $m$ specified in the off-diagonal elements of a migration matrix $M$, where the diagonal elements store the population sizes, $\theta$, of each deme. For $K$ demes, the population size parameter 'theta' is a $K$-tuple, and $m$ is a $(K^2 - K)$-tuple.

```
M = migrationMatrix(theta=[0.1, 0.1], m=[1.0, 1.0]);
g ~ StructuredCoalescent(M=M, n=[15, 15]);
```

In the example above, `migrationMatrix` is a deterministic function and `StructuredCoalescent` is a generative distribution. A stochastic node 'g' stores a gene tree sampled from a two-deme structured coalescent process.

**Skyline coalescent model.** The skyline coalescent model [31] is a coalescent process that models changes in population sizes. This model is characterized having a constant population size for each coalescent interval, with instantaneous changes in population size at some coalescent events.

The following script specifies a Skyline coalescent model with 10 coalescent intervals (11 taxa), with four distinct population sizes.

```
g ~ SkylineCoalescent(theta=[0.1, 0.2, 0.3, 0.4], groupSizes=[4,3,2,1]);
```

Here, 'g' is a stochastic node in the PGM, with its value sampled from the `SkylineCoalescent` generative distribution. Ten coalescent intervals are defined through the 'groupSizes' argument: the first four coalescent intervals will be drawn assuming a 'theta' of 0.1, the next three intervals with 'theta' equal to 0.2, and so on.

**Birth-death models.** Birth-death models are commonly used in macroevolution as sampling distributions for species trees. Models that parameterize the fossilization process can be especially useful, as they allow users to leverage fossil ages as data. When fossil morphological characters have also been scored, total-evidence dating can be carried out [12]. One such tree model is the serially sampled birth-death process [32], whose parameters 'psi' and 'rho' below represent the rate of sampling extinct and extant lineages, respectively:

```
ages = [0.0, 1.0, 2.0, 3.0, 4.0];
tree ~ BirthDeathSerialSampling(lambda=1, mu=0.5, rho=0.1,
psi=1, rootAge=5, ages=ages);
```

Other tree models include the birth-death [33] and fossilized birth-death processes [34], as well as the Yule process [35].

**2.1.9 Substitution models.**   Substitution models consist of continuous-time Markov chains (CTMC) used to model the evolution of discrete characters, such as nucleotides and amino acid residues. LPhy implements a general formulation of a phylogenetic CTMC, known as the GTR model [36], under which several nested models can be specified. The first line below constructs the instantaneous rate matrix Q for an HKY model [37], which is then used to in PhyloCTMC, the generative distribution for the sequence alignment data D:

```
taxa = taxa(names=1:10);
Q = hky(kappa=2.0, freq=[0.2, 0.25, 0.3, 0.25]);
Θ ~ LogNormal(meanlog=3.0, sdlog=1.0);
ψ ~ Coalescent(theta=Θ, taxa=taxa);
D ~ PhyloCTMC(L=200, Q=Q, tree=ψ);
```

Other substitution models can be easily specified by passing different instantaneous transition rate matrices Q to PhyloCTMC, e.g., the matrix of the Jukes-Cantor model [24]:

```
D ~ PhyloCTMC(L=200, Q=jukesCantor(), tree=ψ);
```

For forward simulation PhyloCTMC is used as a generative distribution for a multiple sequence alignment, which is here represented by stochastic node 'D'. When the model is employed for statistical inference, and data D is known, the PhyloCTMC represents the phylogenetic likelihood. More details are discussed in the Data clamping section.

**2.1.10 Evolutionary clock models.**   Molecular clocks are used to model the rate of evolutionary change and how it varies over time. The LPhy language supports strict clock [38, 39], local clock [40] and relaxed clock [41] models. Specifying a clock model is done by generating evolutionary rate values, one per phylogenetic tree branch, and then multiplying those rates by the length of the corresponding branch. The branches of the tree are measured in units of time, effectively scaling the tree to the number of expected substitutions per site.

The simplest clock model is the strict clock, where the evolutionary rate remains constant over the entire tree. Specifying a strict molecular clock can be done by specifying the 'mu' parameter in the PhyloCTMC distribution. The default value for the clock rate 'mu' is 1.0.

```
λ ~ LogNormal(meanlog=3.0, sdlog=1.0);
ψ ~ Yule(lambda=λ, n=16);
D ~ PhyloCTMC(L=200, Q=jukesCantor(), tree=ψ, mu=0.5);
```

More realistic clock models like the uncorrelated relaxed clock model [41] assume the rate for each branch is drawn according to a parametric distribution. For example, a relaxed clock with rates drawn from a log-normal distribution can be constructed as follows:

```
λ ~ LogNormal(meanlog=3.0, sdlog=1.0);
ψ ~ Yule(lambda=λ, n=16);
branchRates ~ LogNormal(sdlog=0.5, meanlog=-0.25, replicates=ψ.branchCount());
D ~ PhyloCTMC(L=200, Q=jukesCantor(), branchRates=branchRates, tree=ψ);
```

Here, 30 rates are drawn independently from a log-normal distribution, and then each is assigned to one of the 30 branches of tree ψ.

**2.1.11 Inference and data clamping.** In addition to simulation, LPhy allows users to use a specified model for inference. We have developed LPhyBEAST which provides support for inference with the BEAST2 engine.

To set up an inferential analysis with LPhy 'data clamping' is performed similar to the Rev language [10]. Data clamping involves associating an observed value with a random variable in the model, which is represented as a stochastic node in the probabilistic graphical model (PGM). By clamping data to a node, the user is informing the inference engine that the value of that particular variable is known and will be conditioned on for the purpose of inference.

In LPhy, data clamping can be accomplished using the 'data block'. This block allows the user to specify the observed values of certain variables in the model, effectively clamping these variables to their observed values during inference. This is useful when working with real data, as it allows the user to incorporate the observed data into the analysis and improve the accuracy of the results.

In LPhy, data clamping can be achieved using the 'data block', for example:

```
data {
    options = {ageDirection="forward", ageRegex="s(\d+)"};
    nexusFilePath = "tutorials/data/RSV2.nex";
    D = readNexus(file=nexusFilePath, options=options);
    codon = D.charset(["3-629\3", "1-629\3", "2-629\3"]);
    n = 3;
    L = [209, 210, 210];
    taxa = D.taxa();
}
model {
    π ~ Dirichlet(replicates=n, conc=[2.0, 2.0, 2.0, 2.0]);
    κ ~ LogNormal(sdlog=0.5, meanlog=1.0, replicates=n);
    r ~ WeightedDirichlet(conc=rep(element=1.0, times=n), weights=L);
    μ ~ LogNormal(meanlog=-5.0, sdlog=1.25);
    Θ ~ LogNormal(meanlog=3.0, sdlog=2.0);
    ψ ~ Coalescent(taxa=taxa, theta=Θ);
    Q = hky(kappa=κ, freq=π, meanRate=r);
    codon ~ PhyloCTMC(L=L, Q=Q, mu=μ, tree=ψ);
}
```

**Example 2**. An LPhy script for phylodynamic analysis of a virus dataset containing Respiratory syncytial virus subgroup A (RSVA) genomic samples [42, 43].

In Example 2, we used a Respiratory syncytial virus subgroup A (RSVA) dataset [42, 43] containing 129 molecular sequences coding for the G protein collected between years 1956 and 2002. We use three partitions corresponding to the codon position, an HKY substitution model [37], coalescent tree prior [25] and a strict molecular clock with a log-normal prior on the mean clock rate. Within the *data* block we clamp the value of 'codon', a stochastic node that appears below inside the *model* block. This is achieved by specifying a data node of the same name (codon) in the data block. In this example the data is vectorized into three codon positions to allow different site models for the different codon positions.

## 2.2 LPhyStudio

Along with the language definition, we introduce LPhy Studio, a Graphical User Interface, GUI, intended for (i) model specification, (ii) PGM graphical and textual display, and (iii) visualization of simulated data. Fig 2 shows a screenshot of LPhyStudio after a simple phylogenetic model was specified. LPhyStudio's features include a scripting console, syntax highlighting, generation of PGMs and natural language text narratives of models with citations, and optionally exporting these PGM and narratives as LaTeX documents. LPhy scripts can be imported using the toolbar or file menu, or created using the scripting console.

## 2.3 LPhy and BEAST2

To facilitate the application of specified models for evolutionary inference, the companion program 'LPhyBEAST' was developed as an interface between LPhy and BEAST2. LPhyBEAST is a command-line tool that takes as input an LPhy script file specifying a model with simulated or observed data, and produces a BEAST2 XML file as output.

# 3 Results

We present key features of the LPhy software using two models from the Phylonco package [27] developed for single-cell data—the GT16 substitution model [44] and the GT16 error model [44]. Starting from an LPhy script, our software generates a text description of the model, a graphical representation of the model as a PGM, and a view of the simulated data shown in Fig 3. Additionally, we also showcase how LPhy can be used to validate the correctness of the BEAST 2 implementation of GT16.

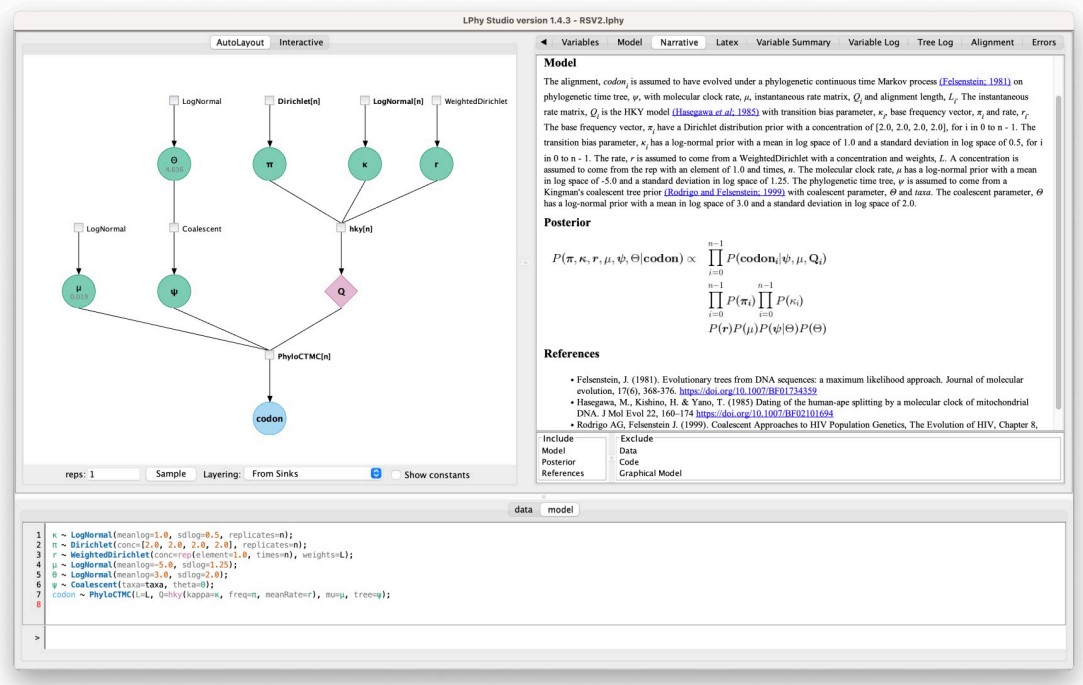

**Fig 2. A screenshot of LPhy Studio showing the probabilistic graphical model on the left panel (constants hidden), and the auto-generated text description of the data and phylogenetic model on the right panel.**

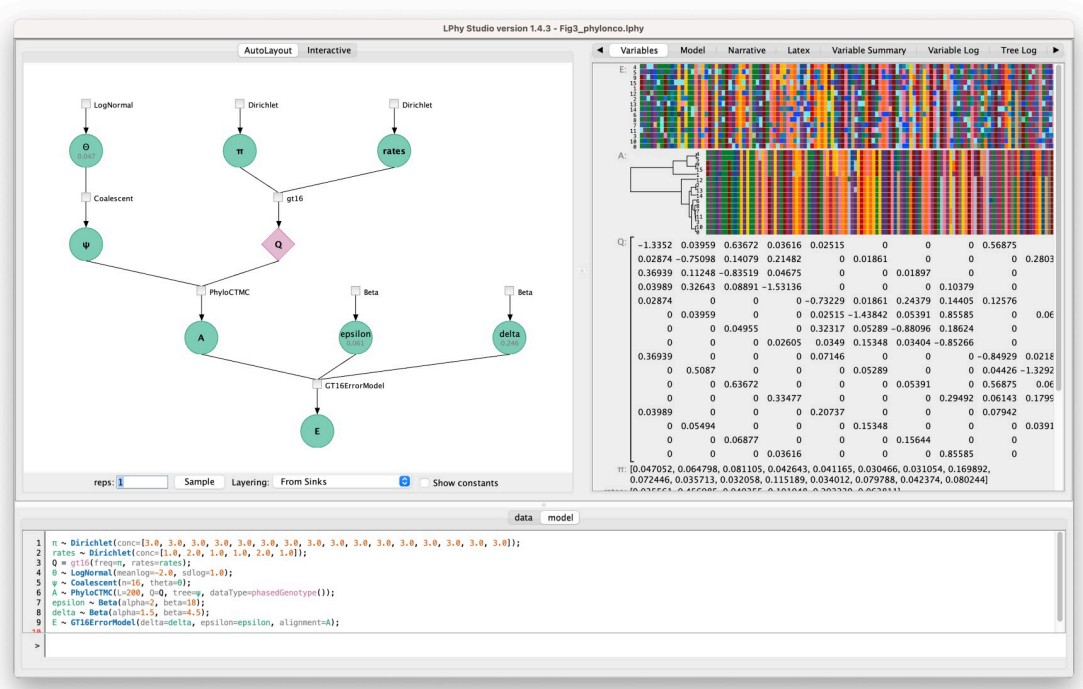

**Fig 3. A screenshot of LPhyStudio showing the GT16 substitution and GT16 error model [27, 44].** The left panel shows the graphical model representation, the right panel shows the simulated tree and diploid nucleotide genotypes, and the bottom panel shows the LPhy script.

### LPhy script

We start from an LPhy script shown in Example 3 which specifies a GT16 substitution model and GT16 error model [44] for simulating single-cell diploid nucleotides with sequencing/amplification error (epsilon) and allelic dropout error (delta). The script defines that 16 sequences 'A' are generated from a GT16 substitution model with rates and genotype frequencies drawn from dirichlet distributions, and a coalescent tree prior with a log-normally distributed theta. The observed noisy sequences 'E' are generated by applying sequencing/amplification error (epsilon), and allelic dropout error (delta) drawn from beta distributions to the sequences 'A'. In the dirichlet generator, the argument 'conc' represents the concentration parameter.

### Natural language description

LPhyStudio can automatically generate a text description of the model as a narrative. The implementation of automated natural language descriptions for phylogenetic models is not new, and similar efforts (albeit for a different scope of phylogenetic methods) can be found in SplitsTree [45] and MEGA4 [46]. The natural language narrative tool we have developed provides a precise starting point for the model description section in a research article that uses Bayesian phylogenetic inference. The LPhy script in Example 3 generates the following narrative.

```
model {
  π  ~ Dirichlet(conc=[3.0, 3.0, 3.0, 3.0,
                       3.0, 3.0, 3.0, 3.0,
                       3.0, 3.0, 3.0, 3.0,
                       3.0, 3.0, 3.0, 3.0]);
  rates ~ Dirichlet(conc=[1.0, 2.0, 1.0, 1.0, 2.0, 1.0]);
  Q = gt16(rates=rates, freq=π);
  Θ ~ LogNormal(meanlog=-2.0, sdlog=1.0);
  ψ ~ Coalescent(n=16, theta=Θ);
  A ~ PhyloCTMC(L=200, Q=Q, dataType=phasedGenotype(), tree=ψ);
  delta ~ Beta(alpha=1.5, beta=4.5);
  epsilon ~ Beta(alpha=2, beta=18);
  E ~ GT16ErrorModel(alignment=A, delta=delta, epsilon=epsilon);
}
```

**Example 3**. An Lphy script defining a GT16 substitution and GT16 error model for diploid single-cell nucleotide data.

The alignment, *E* has an error model [44] with sequencing and amplification error probability, epsilon, allelic dropout probability, delta and genotype alignment, *A*. The genotype alignment, *A* is assumed to have evolved under a phylogenetic continuous time Markov process [47] on phylogenetic time tree, *ψ*, with instantaneous rate matrix, *Q*, an alignment length of 200 and the data type used for simulations. The instantaneous rate matrix, *Q* is the general time-reversible rate matrix on phased genotypes [44] with relative rates, **rates** and base frequencies, *π*. The base frequencies, *π* have a Dirichlet distribution prior with a concentration of [3.0, 3.0, 3.0, 3.0, 3.0, 3.0, 3.0, 3.0, 3.0, 3.0, 3.0, 3.0, 3.0, 3.0, 3.0, 3.0]. The relative rates, **rates** have a Dirichlet distribution prior with a concentration of [1.0, 2.0, 1.0, 1.0, 2.0, 1.0]. The data type used for simulations is the phased genotype data type. The phylogenetic time tree, *ψ* is assumed to come from a Kingman's coalescent tree prior [28] with coalescent parameter, Θ and an n of 16. The coalescent parameter, Θ has a log-normal prior with a mean in log space of -2.0 and a standard deviation in log space of 1.0. The allelic dropout probability, delta has a Beta distribution prior with an alpha of 1.5 and a beta of 4.5. The sequencing and amplification error probability, epsilon has a Beta distribution prior with an alpha of 2 and a beta of 18.

## Model validation

The LPhy framework can be used to verify implementation correctness of new models, in which case they are said to be well-calibrated. Bayesian model validation consists of a series of steps, the first of which is simulation of synthetic data. Recent examples that employ well-calibrated simulation studies within the BEAST2 platform include models for single-cell sequencing errors [27] and correlated continuous traits [48]. By making it possible to simulate complex models and connect to an inference engine (e.g., BEAST 2), LPhy and LPhyBEAST can greatly simplify the validation procedure. Fig 4 presents the validation results for the model described above, when model specification and simulation were performed using LPhy, LPhyBEAST and Phylonco.

## Availability and future directions

LPhyStudio and LPhyBEAST are available on github. This suite of programs is accompanied by a user guide and extensive documentation available via the homepage https://linguaphylo.

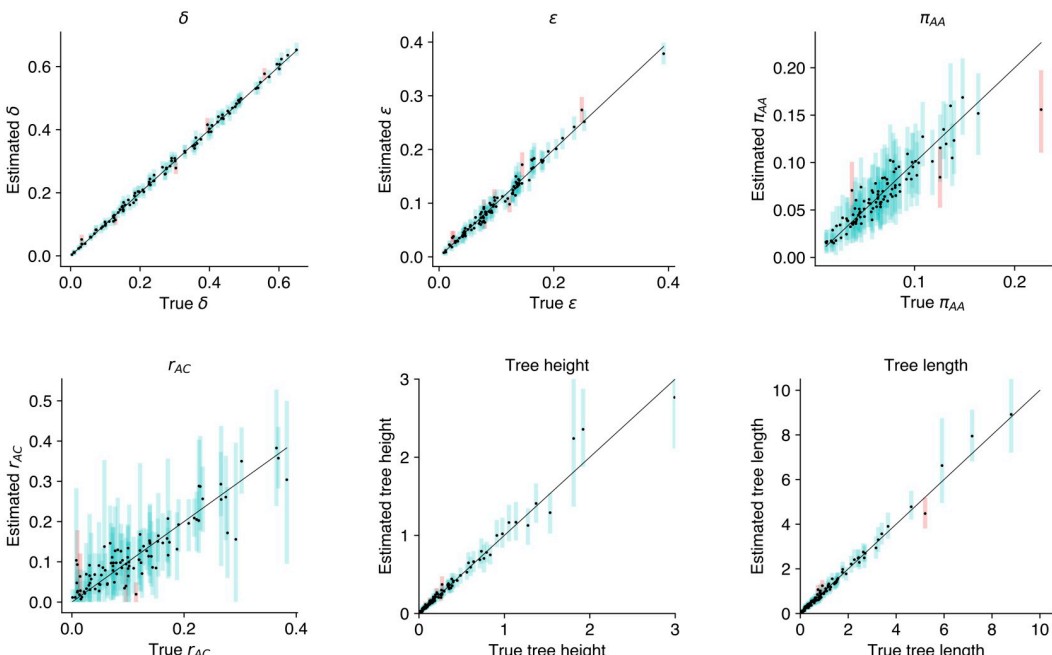

**Fig 4. Model validation for the GT16 substitution model and GT16 error model.** Each plot shows the 95% highest posterior density for model parameters: allelic dropout error $\delta$, sequencing/amplification error $\epsilon$, equilibrium frequency for $\pi_{AA}$, relative rate $r_{AC}$, tree height, and tree length.

github.io/. A growing list of tutorials on the webpage covers common use cases and extension mechanisms. LPhy is designed in a modular fashion, researchers interested in implementing new models within the LPhy language can do so by releasing extension modules that can extend the LPhy application post-deployment.

Although there are many programming languages through which statistical models can be succinctly described (e.g., Stan [20], JAGS [17], BUGS [18, 19]), these languages do not support the unique feature of phylogenetic models—the phylogenetic tree. Phylogenetic trees are complex high-dimensional objects, part discrete, part continuous. There is no bijection between tree space and Euclidean space, so these objects cannot be treated with standard statistical distributions [49]. Hence, specialist software is commonly employed to perform inference involving phylogenetic trees [6–10].

LinguaPhylo differs from existing specialist software in the way it handles model specification. By using vectorization, LinguaPhylo obviates the need for for-loop control flow to describe repetitive structural elements of a model. This feature lowers the risk of syntactic or programming logic mistakes when defining a model relative to a full programming language such as Rev [10]. In its declarative nature, LPhy's language resembles the XML specification adopted by BEAST 2 [7, 8], but shares the central notion of probabilistic graphical models with the Rev language.

LinguaPhylo provides for a form of array programming (vectorization), so that any function or generative distribution can be called with its arguments in vectorized form. In such situations the function or generative distribution is 'broadcast' over each element of the array, which allows for very concise model descriptions.

Future work on integration of LPhy with other popular Bayesian phylogenetic inference tools, such as RevBayes [10], BEAST [6], MrBayes [9] will increase the flexibility of the

framework, and enable easy validation of and comparison between different Bayesian phylogenetic inference engines.

## Supporting information

**S1 Appendix. List of distributions and functions in the LPhy reference implementation.** (PDF)

## Acknowledgments

We thank the New Zealand eScience Infrastructure (NeSI) for access to high-performance computation resources.

## Author Contributions

**Conceptualization:** Alexei J. Drummond.

**Funding acquisition:** Alexei J. Drummond.

**Methodology:** Alexei J. Drummond.

**Project administration:** Alexei J. Drummond.

**Software:** Alexei J. Drummond, Kylie Chen, Fábio K. Mendes, Dong Xie.

**Supervision:** Alexei J. Drummond.

**Validation:** Kylie Chen, Dong Xie.

**Visualization:** Alexei J. Drummond, Dong Xie.

**Writing – original draft:** Alexei J. Drummond, Kylie Chen, Fábio K. Mendes, Dong Xie.

**Writing – review & editing:** Alexei J. Drummond, Kylie Chen, Dong Xie.

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
