## [Decision Letter · Decision Letter 0]

31 Jan 2023

Dear Drummond,

Thank you very much for submitting your manuscript "LinguaPhylo: a probabilistic model specification language for reproducible phylogenetic analyses" for consideration at PLOS Computational Biology.

As with all papers reviewed by the journal, your manuscript was reviewed by members of the editorial board and by several independent reviewers. In light of the reviews (below this email), we would like to invite the resubmission of a significantly-revised version that takes into account the reviewers' comments.

Three experts have commented on the new manuscript with various levels of enthusiasm. Some reviewers were not impressed or found the work to be narrower. In all cases, however, there was support for the topic and the initial prototype, but some concerns about the general usefulness of this tool. I think it is worthwhile to develop such languages/tools, but whether they will become broadly adopted is something one can only decide right away. Still, I feel that if this can be used extensively, even for one major widely-used software, then it will be an important advance. For example, my group developed a smart Caption Expert tool for MEGA in 2007 which is widely used (see 2007 article https://doi.org/10.1093/molbev/msm092), which is primarily intended to give users an idea of what they have done. As reviewer 1 also points out that extracting the textual description of the analysis, including citations, has been around in SplitsTree also (since 2018). Many researchers use such information in their Methods sections and figure legends. So, I think these efforts should be acknowledged. Therefore, I have a significant appreciation for a tool that can provide natural language narratives. This tool will fulfill that need for at least the intended software. But, the reviewers make many important suggestions, which would greatly improve the quality of this manuscript. It includes extending the language to deal with the scenarios noted by the reviewers and dealing with the package issues to make it more widely useful. Also, the manuscript needs to be revised extensively to accommodate all other comments made by the reviewers, as much as possible.

We cannot make any decision about publication until we have seen the revised manuscript and your response to the reviewers' comments. Your revised manuscript is also likely to be sent to reviewers for further evaluation.

Sincerely,

Sudhir Kumar

Guest Editor

PLOS Computational Biology

Thomas Leitner

Section Editor

PLOS Computational Biology

Three experts have commented on the new manuscript with various levels of enthusiasm. Some reviewers were not impressed or found the work to be narrower. In all cases, however, there was support for the topic and the initial prototype, but some concerns about the general usefulness of this tool. I think it is worthwhile to develop such languages/tools, but whether they will become broadly adopted is something one can only decide right away. Still, I feel that if this can be used extensively, even for one major widely-used software, then it will be an important advance. For example, my group developed a smart Caption Expert tool for MEGA in 2007 which is widely used (see 2007 article https://doi.org/10.1093/molbev/msm092), which is primarily intended to give users an idea of what they have done. As reviewer 1 also points out that extracting the textual description of the analysis, including citations, has been around in SplitsTree also (since 2018). Many researchers use such information in their Methods sections and figure legends. So, I think these efforts should be acknowledged. Therefore, I have a significant appreciation for a tool that can provide natural language narratives. This tool will fulfill that need for at least the intended software. But, the reviewers make many important suggestions, which would greatly improve the quality of this manuscript. It includes extending the language to deal with the scenarios noted by the reviewers and dealing with the package issues to make it more widely useful. Also, the manuscript needs to be revised extensively to accommodate all other comments made by the reviewers, as much as possible.

Reviewer's Responses to Questions

**Comments to the Authors:**

Reviewer #1: This paper presents a language for describing models and data in the context of statistical modeling of phylogenetic genetics. At present, the framework can be used for two purposes: - to simulate data and to setup input files for BEAST2. The authors indicate that they plan to extend the program to generate input files for other Bayesian phylogenetic frameworks in the future.

The main content of the paper is to demonstrate how one would use the language to setup different models, by example. It also illustrates some of the features of an interactive program called LPhy Studio.

I believe that the specification language, and, more importantly, the underlying engine that produces output based on the inputted specification, are value contributions to phylogenetics.

I assume that the code for generating data according to the different models already existed and so the main technical contribution here was probably to implement a parser for the specification language.

However, I do have some questions.

First, I think that the paper should say something about language features, e.g.: what language is the syntax modeled on, do variables have types? Are the methods overloaded? When does the Studio flag errors, does it check syntax before you execute or only during execution? The examples use Greek letters. Is that required? The section on "Language Features" should contain more on this.

Why is this not a Python package? I believe that scripting is best done using a scripting language. I don't see why the parser and executor are written in Java. Even if there are good reasons for providing this in a Java framework, I suggest that providing a Python (or R?) package with the same functionality, as this would make this much more widely usable.

The paper mentions a command line program for converting scripts into Beast2 configurations. There should also be a commandline program for running a script to create simulated data, too.

I took a closer look at LPhy Studio and came across several issues. First, there is no installer, so one has to download a bunch of files and then run a jar from the commandline line. That is not state-of-the-art. Second, the program is hard to use because it does not contain the standard menus and menu items. In particular, there is no Edit menu, and therefore no cut, copy, paste, delete, undo and redo menu items. The File menu doesn't contain a New... menu item and so program appears to only support one open document at a time. There is no print menu item. The File menu contains a Tutorials submenu that doesn't belong there.

The interface itself is bare bones. There is no tool bar with the usual useful items. All content is organized into two tab panes and a text area, and the tab panes are not splittable, and so only a limited amount of content can be viewed together, even on a big screen. The model text editor appears to be a single line-editor. There does not appear to be any (easy) way to change something that has been entered (no undo redo, and no editing of the model script?). The single-line editor doesn't have a history buffer.

Looking into the code of LPhy Studio reveals that the UI is implemented using AWT and Swing. These technologies are severely outdated and explain why the UI of LPhy Studio feels so minimalistic and dated. However,

the website mentions that start-of-the-art technologies are used?

The idea of representing phylogenetic analyses as graphs and generating a methods section ("narrative") from the graph to provide a precise, human readable description of the analysis, including citations, should

be credited to Daniel Huson. This idea was presented by him in 2018 at a phylogenetic meeting in Dunedin. This presentation contained a figure showing a split window, with a graph representing a phylogenetic

analysis above and a description of the analysis, together with all relevant citations below, very similar to Figure 2 of the manuscript. Alexei Drummond was present at the meeting and commented favorably on the idea at the time.

The current wording of the manuscript might suggest that this is a new idea due to the authors and so it would be prudent to add an appropriate sentence to avoid that impression, something like this:

"The idea of explicitly modeling the for provenance of a phylogenetic analysis as a graph, and using this graph to generate a textual description of the analysis, together with all relevant citations, was introduced in 2018 with version of 5 of SplitsTree (Huson and Bryant, 2006)."

Reviewer #2: The manuscript "LinguaPhylo..." attempts to solve the problem of simplifying descriptions of phylogenetic models to increase both repeatability and consistent description for methods. They do so by developing a simple model language, superficially similar to what one might find in other packages, that can theoretically stand independently of a particular software package, although currently the only integration is for BEAST 2.

The concept is straight-forward enough, although the practicality and complexity remains to be seen through actual use and, while adoption by other related software packages such as RevBayes or BEAST would likely be necessary for it to gain general traction.

In additional to the LPhy language, two tools are mentioned, an independent GUI, LPhyStudio which is designed to create graphical representations of models as well as "methods ready" descriptions of the model (not dissimilar to the "figure caption generation" seen in other packages such as MEGA), as well as LPhyBeast which is designed to create an Beast2 input XML from a LPhy model.

A few general issues strike me, none of them major.

1. The authors abbreviate "LinguaPhylo" as "LPhy" and state it is pronounced "el-fee" (p. 2). That is a losing battle as absolutely everyone is going to pronounce it "el-fai" given that it is "fai-logenetics" and not "fee-logenetics".

2. I am not an expert but the colors used in the figures, generated by LPhyStudio, including not just graphical representations but even color-coding of the LPhy text as shown throughout the manuscript, is very likely not color-blind friendly as it heavily makes use of green and red (reddish-appearing) colors (generally speaking, lime green (#00FF00) is not a particularly good color for text anyway as it is too bright on a white background)

3. I'm sure there is a logic to it, but I cannot quite make out within the LPhy language when a parameter is set with an = vs. a ~. I think it might be deterministic vs. stochastic, but the authors never specify this (if correct, I would suggest p. 3 where these types of generators are discussed).

4. The very first example of the language (Listing 1) is meant to be simple, but specifies the taxa "names" as the numbers 1 through 10. This inadvertently makes it appear that the language requires numbered taxa rather than named taxa, which I do not believe to be true.

5. There is at least one typo, p. 3 "distrivutions"

6. While I know what it means, the authors use "i.i.d." at least twice without ever defining it. It should be defined at the time of first use

7. The authors do not explain that the parameter "conc" is an abbreviation for "concentration" other than secondarily through the automatically generated natural text description toward the end of the manuscript on p. 10.

8. The natural text narrative in the examples have some oddities to them, but to be fair I'm not sure there is a way of fully auto-generating this sort of text in a perfect manner. Just as some examples, in the text shown on p. 10,

** The second sentence ends with "and a dataType" which makes no logical sense.

** The last two sentences begin with "The delta has a..." and "The epsilon has a..." when grammatical logic would suggest not starting either sentence with the word "The" or adding an additional explanation of what delta and epsilon refer to, e.g., "The delta of the error model..." and "The epsilon of the error model..." or something to that effect.

I could nitpick other parts of the output, but as I said, I'm not sure it is possible to make a perfect one-stop solution to this natural text problem.

Reviewer #3: This manuscript presents LinguaPhylo (LPhy), a language for easily defining models for BEAST 2 analysis. LPhy is much simpler than the normal (complicated) XML format, a welcome feature for BEAST 2 users. It is also coupled with LPhyStudio, a graphical interface, and LPhyBEAST, a command-line tool to convert the LPhy to the XML file.

Overall, these tools are certainly worthwhile to the BEAST community. However, I failed to understand how useful it is to other Bayesian software or phylogenetic software, in general. If LPhy is presented as a “language” (as the title and abstract suggest), then there should be information how it can be generally adopted, outside of BEAST. This is currently missing.

For phylogenomic inference, mixture models are now widely used, especially quite important to address deep or controversial phylogenetic questions; and also useful for more recent questions such as incomplete lineage sorting. I don’t find any information how to specify mixture models. (e.g, can I use the CAT model in PhyloBayes?).

Is LPhy format case-sensitive?

And is it possible to adopt LPhy for maximum likelihood analysis? This can be seen as a subset of the current language (i.e., excluding the priors). So I don’t see why not. One can even think about a more lightweight version of LPhy, which will benefit the ML community.

**Have the authors made all data and (if applicable) computational code underlying the findings in their manuscript fully available?**

Reviewer #1: Yes

Reviewer #2: Yes

Reviewer #3: Yes

PLOS authors have the option to publish the peer review history of their article (what does this mean?). If published, this will include your full peer review and any attached files.

Reviewer #1: **Yes: **Daniel Huson

Reviewer #2: No

Reviewer #3: No
---

## [Editor Report · Decision Letter 1]

30 May 2023

Dear Dr. Drummond,

We are pleased to inform you that your manuscript 'LinguaPhylo: a probabilistic model specification language for reproducible phylogenetic analyses' has been provisionally accepted for publication in PLOS Computational Biology.

Should you, your institution's press office or the journal office choose to press release your paper, you will automatically be opted out of early publication. We ask that you notify us now if you or your institution is planning to press release the article. All press must be coordinated with PLOS.

Best regards,

Sudhir Kumar

Guest Editor

PLOS Computational Biology

Thomas Leitner

Section Editor

PLOS Computational Biology

The authors have considered reviewers' comments carefully and made updates that have improved the manuscript.

---

## [Editor Report · Acceptance letter]

13 Jul 2023

PCOMPBIOL-D-22-01241R1 

LinguaPhylo: a probabilistic model specification language for reproducible phylogenetic analyses

Dear Dr Drummond,

I am pleased to inform you that your manuscript has been formally accepted for publication in PLOS Computational Biology. Your manuscript is now with our production department and you will be notified of the publication date in due course.

With kind regards,

Bernadett Koltai
